

# Runoff Evaluation in an Earth System Land Model for Permafrost Regions

Xiang Huang[1], Yu Zhang[1*], Bo Gao[2*], Charles J. Abolt[1,3], Ryan L. Crumley[1], Cansu Demir[1], Richard P. Fiorella[1], Bob Busey[4], Bob Bolton[2], Scott L. Painter[2], and Katrina E. Bennett[1,5]

[1]Earth and Environmental Science Division, Los Alamos National Laboratory, Los Alamos, NM, USA
[2]Environmental Sciences Division, Oak Ridge National Laboratory, Oak Ridge, TN, USA
[3]Science and Analytics Team, The Freshwater Trust, Portland, OR, USA
[4]University of Alaska Fairbanks, Fairbanks, AK, USA
[5]Analytics, Intelligence, and Technology Division, Los Alamos National Laboratory, Los Alamos, NM, USA
*Second and third author did equal work on this manuscript

*Correspondence to*: Xiang Huang (huangx07@gmail.com) and Katrina E. Bennett (kbennett@lanl.gov)

**Abstract.** Modeling of hydrological runoff is essential for accurately capturing spatiotemporal feedbacks within the land–atmosphere system, particularly in sensitive regions such as permafrost landscapes. However, substantial uncertainties persist in the terrestrial runoff parameterization schemes used in Earth system and land surface models. This is particularly

true in permafrost regions, where landscape heterogeneity is high and reliable observational data are scarce. In this study, we evaluate the performance of runoff parameterization schemes in the Energy Exascale Earth System Model (E3SM) land model (ELM). Our proposed framework leverages simulation results from the Advanced Terrestrial Simulator (ATS), which is a physics-rich integrated surface/subsurface hydrologic model that has been successfully evaluated previously in Arctic tundra regions. We used ATS to simulate runoff from 22 representative hillslopes in the Sagavanirktok River basin, located

on the North Slope of Alaska, then compared the output with ELM's parameterized representation of total runoff. Results show that 1) ELM's total runoff was the same order of magnitude as the ATS simulations, and both models were similarly variable over time; 2) minor adjustments to coefficients in ELM's runoff parameterization improved the match between the ATS simulation and ELM's parameterized representation of annual and seasonal total runoff; 3) overall, runoff responses in ATS and ELM are more similar in flat hillslope environments compared to steep hillslopes; and 4) shallower active layer

thicknesses and higher precipitation simulations resulted in lower correlations between the two models due to greater total runoff. By incorporating the optimized runoff coefficients from the Sagavanirktok River basin into ELM, the simulated total runoff better matched the streamflow observations at a small watershed located on the Seward Peninsula of Alaska. Our findings revealed important insights into the effectiveness of runoff parameterizations in land surface models and pathways for improving runoff coefficients in typical Arctic regions.




## 1 Introduction

Runoff parameterization schemes play a critical role in the accuracy of Earth system models, particularly in sensitive environments such as high latitude permafrost regions. These areas are increasingly vulnerable to climate variability and the hydrological responses associated with warming temperatures can have profound implications for ecosystems and water resources (Bring et al., 2016; Yang & Kane, 2020). The interplay between hydrology and climate dynamics in permafrost zones is complex because conditions such as vegetation, snow, soil wetness, ground ice content, and biogeochemical
activities vary significantly over small spatial extents (Holmes et al., 2013; Bennett et al., 2022).

Despite the importance of runoff in accurately modeling ecosystem dynamics, considerable uncertainties remain in the parameterization schemes employed by land surface models. The accuracy of one process often depends on the scheme chosen for another, creating interdependencies that can complicate model accuracy. In permafrost regions, the presence of ice-rich permafrost can disrupt water infiltration processes, leading to increased surface runoff and altered drainage patterns
(Kuchment et al., 2000; Walvoord & Kurylyk, 2016; Bennett et al., 2023). These heterogeneous conditions complicate efforts to accurately represent hydrological dynamics and highlight the necessity for improved modeling techniques. The scarcity of observational data and unmeasurable model parameters exacerbate these challenges, resulting in significant discrepancies between model outputs and real-world hydrological behavior (Bring et al., 2016; Clark et al., 2015, 2017). Addressing these uncertainties is essential for developing reliable predictive models that can support resource management
and conservation efforts within rapidly changing Arctic ecosystems (Harp et al. 2016).

Recent land model intercomparison projects (e.g., Boone et al., 2004; Lawrence et al., 2016; Collier et al., 2018; Mwanthi et al., 2024) have summarized various implementations of runoff schemes, ranging from simple bucket models to more advanced topography-based runoff models. These studies highlight significant variability in lateral surface runoff and subsurface runoff (baseflow) among different land models. Clark et al. (2015) emphasized the need to integrate groundwater-
surface water interactions in Earth system models, while Maxwell et al. (2014) demonstrated the benefits of coupling surface and subsurface models for better predictions in complex landscapes. By improving soil freeze-thaw processes and incorporating soil heterogeneity, Liang & Xie (2001) and Swenson et al. (2012) achieved better runoff alignment with observed streamflow. Fan et al. (2019) identified lateral water flow as crucial runoff component for the water cycle in the Arctic, with additional studies highlighting significant uncertainties in runoff parameterization schemes in high-latitude cold
regions (e.g., Zheng et al., 2017; Hou et al., 2023; Abdelhamed et al., 2024). These efforts collectively highlight the pressing need to refine hydrological runoff simulations to improve predictions particularly in permafrost regions as climate change intensifies.

Many previous studies have evaluated runoff parameterization by comparing different schemes against streamflow observations at large scales and coarse resolutions (e.g., Niu et al., 2007; Li et al., 2011; Swenson et al., 2012; Zheng et al.,
2017; Li et al., 2024), however high-quality streamflow data that can be used to validate runoff production are difficult to obtain in permafrost regions. This highlights the need for a more cost-effective and flexible framework to rapidly evaluate



parameterization effectiveness using alternative approaches, for example, leveraging simulations from robust computational tools for physics-rich permafrost thermal hydrology processes. The permafrost thermal hydrology capability (Painter et al. 2016) in the Advanced Terrestrial Simulator (ATS) (Coon et al., 2020) has emerged as a valuable tool in this regard. ATS has been successfully compared to snow depth, supra-permafrost water table depth, and vertical profiles of soil temperatures (Atchley et al. 2015; Harp et al. 2016; Jan et al. 2020) and to catchment-scale evapotranspiration and runoff (Painter et al, 2023) in continuous permafrost regions. ATS's permafrost thermal hydrology capabilities have been used in a variety of modeling studies (e.g., Atchley et al. 2016; Sjöberg et al., 2016; Jafarov et al. 2018; Abolt et al., 2020; Jan & Painter., 2020; Hamm & Frampton, 2021; Painter et al. 2023)

This study aims to evaluate and improve the parameterization of runoff processes in the Department of Energy's Energy Exascale Earth System Model (E3SM) Land Model (ELM) (e.g., Oleson et al., 2013; Bisht et al., 2018; Xu et al., 2022; Shi et al., 2024) using detailed simulations from ATS. We quantitatively assess ELM's runoff parameterization, focusing on total runoff rather than individual components separately. By adopting a total water mass balance perspective, this approach provides insights into the strengths and limitations of ELM's runoff schemes, ultimately enhancing its predictive capabilities in Arctic environments. Additionally, it offers a comprehensive understanding of how landscape features and thermal hydrological processes interact in permafrost regions.

## 2. Model description

### 2.1. ELM runoff parameterization schemes

The runoff parameterization within ELM is designed to represent how water moves across the land surface between grid cells and is influenced by numerous factors, including soil moisture, topography, vegetation cover, etc. ELM's runoff scheme (ported from CLM v4.5, Oleason et al., 2013) is based on a simple TOPMODEL-based concept with a simplified topography representation (Niu et al., 2005). The runoff in ELM is partitioned into surface and subsurface flows, both of which are assumed to be related to water storage, vertical infiltration, and groundwater-soil water interactions (Beven & Kirkby, 1979; Niu et al., 2005, 2007). There are more than twenty runoff components (variables) defined in ELM, but basically, they can be categorized into three groups in a simulation with fixed land use: i) surface runoff, ii) subsurface runoff, and iii) runoff from overland water bodies like wetlands, lakes, and glaciers. The top 10 layers are considered soil to a depth of ~3.8 m and are hydrologically and biogeochemical active. The remaining 5 ground layers in each column are considered to be dry bedrock that extend to a depth of ~42.1 m. Here, we only explicitly list the key equations representing the crucial components of the former two groups representing for surface/subsurface runoff; for a more detailed description of the underlying physics and complete formulations, readers are referred to existing literature (Oleson et al., 2013; Niu et al., 2005; Bisht et al., 2018; Liao et al., 2024).





Surface runoff is composed of two components: i) outflow from the saturated portion of a grid cell with excess water, $q_{over}$, and ii) outflow from surface water storage such as a pond, $q_{h2osfc}$. The first term is written as:

$$q_{over} = q_{liq,0} f_{max} \exp\left(-0.5 f_{over} z_{\nabla,perch}\right) \tag{1}$$

where $q_{liq,0}$ is the sum of liquid precipitation reaching the ground and melt water from snow (kg·m⁻²·s⁻¹); $f_{max}$ is the ratio of the area that has higher compound topographic index (CTI) values than the mean CTI value of the grid cell, with a consideration of geomorphological features; $f_{over}$ is a decay factor that is often calibrated using the recession curve of the observed hydrograph, taken as 0.5 m⁻¹; $z_{\nabla,perch}$ is the perched groundwater table depth (m) within the thawed soil layers. The second term is formulated as:

$$q_{h2osfc} = k_{h2osfc} f_{connected} \left(W_{sfc} - W_c\right) \frac{1}{\Delta t} \tag{2}$$

where the storage coefficient $k_{h2osfc} = \sin(\beta)$ is a function of grid cell mean topographic slope $\beta$ (in radians); $f_{connected}$ is the fraction of the inundated portion of the interconnected grid cell, calculated as $f_{connected} = (f_{h2osfc} - 0.5)^{0.14}$, if $f_{h2osfc}$ is greater than 0, otherwise equal to 0, where $f_{h2osfc}$ is the fraction of the area that is inundated. $W_{sfc}$ represents surface storage water (kg·m⁻²), determined by the surface-inundated fraction $f_{h2osfc}$, the slope $\beta$, the ponded water height, and

microtopographic features. $W_c$ is the amount of surface water present when $f_{h2osfc} = 0.5$; and $\Delta t$ is the model time step.

Subsurface runoff is also composed of two components: i) drainage in the frozen soils where the groundwater table remained dynamic under partially frozen conditions, $q_{drain}$, and ii) drainage from the thawed active layer, $q_{drain,perch}$. The first term is based on the following exponential relationship:

$$q_{drain} = 10^{\Theta_{ice}} \cdot 10\sin(\beta) \cdot \exp\left(-2.5 z_{\nabla}\right) \tag{3}$$

where $\Theta_{ice}$ is an exponent of the ice impedance factor. It is calculated as $\Theta_{ice} = -6\left(\sum_{i=jwt}^{N_{levsoi}} S_{ice,i} \Delta z_i \middle/ \sum_{i=jwt}^{N_{levsoi}} \Delta z_i\right)$, where

$S_{ice,i}$ is the saturation degree of ice in soil layer $i$; $\Delta z_i$ is the layer thickness; $jwt$ is the index of the layer directly above the water table; and $N_{levsoi} = 15$ refers to the total number of soil layers. $z_{\nabla}$ is the groundwater table depth (m), which is initialized at five meters below the soil column (8.8 m). It should be noted that for continuous permafrost or frozen soil, its





drainage is equal to zero or tiny values, and here the last term in Eq. (3) is reduced to a very small value, i.e., $2.8\times10^{-10}$. The

second term refers to the lateral drainage from the perched saturated zone between layers $N_{perch}$ and $N_{frost}$, written as:

$$q_{drain,perch} = 10^{-5}\sin(\beta)\left(\sum_{i=N_{perch}}^{i=N_{frost}}10^{-6\left(\frac{S_{ice,i}+S_{ice,i+1}}{2}\right)}k_{sat,i}\Delta z_i\right)\left(z_{frost}-z_{\nabla,perch}\right) \tag{4}$$

where $k_{sat,i}$ is soil hydraulic conductivity (m·s$^{-1}$) at saturated unfrozen status in soil layer $i$. $z_{frost}$ is the frost table defined

as the shallowest frozen layers having an unfrozen layer above it (m). $z_{\nabla,perch}$ is the perched groundwater table depth (m)

within the thawed layers above icy permafrost ground.

In this study, the total runoff from ELM is calculated as the sum of above four runoff components, expressed as

$q_{total} = q_{drain} + q_{drain,perch} + q_{over} + q_{h2osfc}$ (with units in mass water flux, kg·m$^{-2}$·s$^{-1}$).

## 2.2. ATS runoff generation schemes

ATS solves integrated surface/subsurface flow in complex topographic landscape with complex soil structures, which can

capture a wide array of processes and their interactions to produce a holistic system understanding of a system (Painter et al.,

2016; Coon et al., 2020; Gao & Coon, 2022). As a physics-rich hydrological model, ATS uses physically based

representations for surface runoff, subsurface runoff, and river routing. Here, only the key governing equations are

presented.

The subsurface variably saturated flow is based on the Richards equation with phase change to solve the conservation of

water mass, written as:


$$\frac{\partial}{\partial t}\left[\phi\left(\omega_g m_g s_g + m_l s_l + m_i s_i\right)\right] = -\nabla\cdot\left(m_l \boldsymbol{q}_l\right) + Q_w \tag{5}$$

where $\phi$ is porosity; the subscripts $g$, $l$, and $i$ refer to the gas, liquid, and ice phases; $\omega$ is the gaseous mole fraction

(mol·mol$^{-1}$) referring to a molar fraction of water vapor within all gas in the pore space; $m$ is the molar density of a

particular phase (mol·m$^{-3}$); s is saturation ($s_g + s_l + s_i = 1$); and $Q_w$ refers to sources and sinks (mol·s$^{-1}$). The Darcy

velocity (m·s$^{-1}$) is presented as $\boldsymbol{q}_l = -k_{int}k_{rl}/\mu_l\left(\nabla P_l + \rho_l \boldsymbol{g}\nabla z\right)$, where $k_{int}$ is intrinsic permeability (m$^2$), $k_{rl}$ is relative

permeability, $\mu_l$ is dynamic viscosity (Pa·s), $P_l$ is pressure head (Pa), $\rho_l$ is water density (kg·m$^{-3}$), $\boldsymbol{g}$ is the gravitational

acceleration (m·s$^{-2}$), and z is the vertical elevation (m). The vapor pressure in the pore space is assumed to be in equilibrium

with the liquid phase above the freezing temperature and in equilibrium with the ice phase below freezing. The

parameterizations and constitutive relationships, such as the van Genuchten soil water retention curve and water-ice phase



transition functions are omitted here. The conservation of energy in the subsurface assumes local thermal equilibrium among

the ice, liquid, gas, and solid grains, presented as:

$$\frac{\partial}{\partial t}\left[\sum_{j=l,g,i}\phi m_j s_j u_j +(1-\phi)C_e T\right] = -\nabla\cdot\left(m_l h_l \boldsymbol{q}_l\right)+\nabla\cdot\left(\lambda_e \nabla T\right)+Q_E \tag{6}$$

where $T$ is the temperature (K); $u$ is the is the specific internal energy (J·mol⁻¹); $h$ is the specific enthalpy (J·mol⁻¹); $C_e$

and $\lambda_e$ are the equivalent heat capacity (J·m⁻³·K⁻¹) and thermal conductivity (W·m⁻¹·K⁻¹) of the soil composite (liquid, ice,

gas, and solid grains); $Q_E$ is the thermal energy sources and sinks (W·m⁻³).

The thermal surface flow with phase change is governed by three core equations (Painter et al., 2016): the mass balance for

water in the liquid and ice phases, a diffusion wave approximation for surface flow extended to include an immobile ice

phase, and the energy balance equation. The effects of surface water freezing and thawing are incorporated through a liquid–

ice partitioning factor, or unfrozen fraction $\chi$, which depends on the surface water temperature and varies smoothly from 0

to 1 as the temperature rises through the freezing point. These governing equations are expressed as follows:


$$\frac{\partial}{\partial t}\left[\delta_w \chi m_l +\delta_w (1-\chi)m_i\right]+\nabla\cdot\left(\delta_w \chi m_l \mathbf{U}_w\right)=Q_{ss} \tag{7}$$

$$\mathbf{U}_w = -\frac{(\chi\delta_w)^{2/3}}{n_{mann}\left(\|\nabla Z_s\|+\varepsilon\right)^{1/2}}\nabla\left(Z_s+\delta_w\right) \tag{8}$$

$$\frac{\partial}{\partial t}\left[\delta_w \chi m_l u_l +\delta_w (1-\chi)m_i u_i\right]+\nabla\cdot\left(\delta_w \chi m_l \mathbf{U}_w h_l\right)-\nabla\cdot\left[\delta_w\left(\chi\kappa_l +(1-\chi)\kappa_i\right)\nabla T_s\right]=Q_{ess} \tag{9}$$

where $\mathbf{U}_w$ is the surface flow velocity (m·s⁻¹); $Q_{ss}$ and $Q_{ess}$ are the mass (mol·m⁻²·s⁻¹) and energy (W·m⁻²) source/sink

terms, respectively; $\delta_w$ is ponded depth (m); $n_{mann}$ is Manning's coefficient (s·m⁻¹/³); $\varepsilon$ is a small positive regularization

(m) to keep the equations non-singular in regions with zero slope ratio. The ponded depth and surface elevation $Z_s$ are

defined in two dimensions (x-y) and the vector operators are to be interpreted accordingly.

The land surface energy is calculated either at the surface of a snowpack or ponded water, presented as:

$$(1-\alpha)Q_{sw}^{in}+Q_{lw}^{in}+Q_{lw}^{out}+Q_h(T_s)+Q_e(T_s)+Q_c(T_s)=0 \tag{10}$$

where $\alpha$ is surface albedo and $T_s$ is the surface temperature (K); $Q_{sw}^{in}$ is incoming shortwave radiation (W·m⁻²); $Q_{lw}^{in}$ is

incoming longwave radiation (W·m-2); $Q_{lw}^{out}$ is outgoing longwave radiation (W·m⁻²); $Q_h$ is the sensible heat flux (W·m⁻²);

$Q_e$ is the latent heat flux (W·m⁻²); and $Q_c$ is the conductive heat flux (W·m⁻²).



The above Eqs (5)–(10) represent the integrated surface–subsurface thermal hydrological processes. Continuity of primary scalar fields and fluxes (e.g., pressure, temperature, and water content) is enforced across the surface–subsurface interface. The fully coupled system is solved simultaneously to capture key hydrological dynamics, including freeze–thaw transitions
and energy–water exchanges, enabling the generation of physically consistent hydrological outputs. Here, we consider the cumulative discharge (mol·s$^{-1}$) at the downstream outlet as the total runoff from the simulation domain.

## 3. Methodology

### 3.1. Study areas and model implementations

The first study area is located in the Sagavanirktok (Sag) River basin, located on the North Slope of Alaska (Figure 1a).
Meteorological forcing data for this region are sourced from the Daymet version 4 dataset (Thornton et al., 2020). Temperatures range from −25°C in January to 15°C in July, with approximately half of the annual precipitation occurring as snowfall from September through April, while summer rainfall contributes around 50 % of the total precipitation. The soil properties used in this study are based on previous modeling efforts and include two primary layers: an organic-rich surface layer (10–30 cm thick) composed of mosses, peats, and organic-rich soils, and the underlying mineral soil layer. For further
details on ATS mesh generation, boundary conditions, model setup, and initialization, refer to Jan et al. (2020), Gao & Coon (2022), and Coon et al. (2022). In this study, we randomly selected 22 hillslopes, each parameterized based on sub-catchments within the Sag River basin. These hillslopes span a range of slopes and drainage areas, providing a basis for sensitivity analysis and model development. The 2D variable width hillslopes were derived by downscaling the corresponding 3D sub-catchments through a series of parameterization steps.

The second study area is the Teller watershed, a 2.5 km² drainage basin located approximately 27 miles from Nome on Teller Highway, located on the Seward Peninsula of Alaska (hereafter referred to as Teller27, Figure 1b). Compared to the cold and dry climate of the Sag River basin, the Teller27 site experiences a warmer and wetter climate. The Sag site receives over twice the annual snowfall of Teller27, while the Teller27 site is 7–8°C warmer on average (Gao & Coon, 2022). The Teller27 site was used to evaluate the adjusted runoff coefficients derived from ATS-calibrated simulations in the Sag River
basin. The Sag coefficients were implemented into ELM's source code and tested for transferability by applying them in ELM simulations at Teller27. Unlike the Sag River analysis, no ATS simulations were used at Teller27; instead, ELM performance was evaluated directly against observed streamflow data.

The ELM model was implemented at 0.5-degree resolution of subgrid-scale surface/land use input parameters and driven by ERA5-Land meteorological forcings (Muñoz-Sabater et al., 2021). We used the Offline Land Model Testbed (OLMT) to
standardize ELM case setup and model spin-up (e.g., Sinha et al., 2023). Model spin-up proceeds through two phases after Thornton & Rosenbloom (2005): the first phase features accelerated biogeochemical cycling while the second phase uses standard biogeochemical reaction rates. These spin-up phases are run for 260 and 200 years, respectively, to ensure





vegetation and biogeochemistry has approached a steady-state condition before beginning a transient run that spans 1850-2024.

Streamflow measurements were collected at the Teller27 watershed river outlet (Busey et al., 2019). For additional climate, snow, subsurface properties, and permafrost at the Teller27 site, refer to Bennett et al. (2022), Jafarov et al. (2018), Léger et al. (2019), Thaler et al. (2023), and Wang et al. (2024). To evaluate the influence of soil property parameters on ELM-simulated runoff, a series of parameter sensitivity analyses was conducted. In each simulation, one parameter was perturbed by ±50 % from its default average value (ELM's surface and land use files are extracted from the global 0.5-degree

resolution files), while the values of all other parameters were fixed.

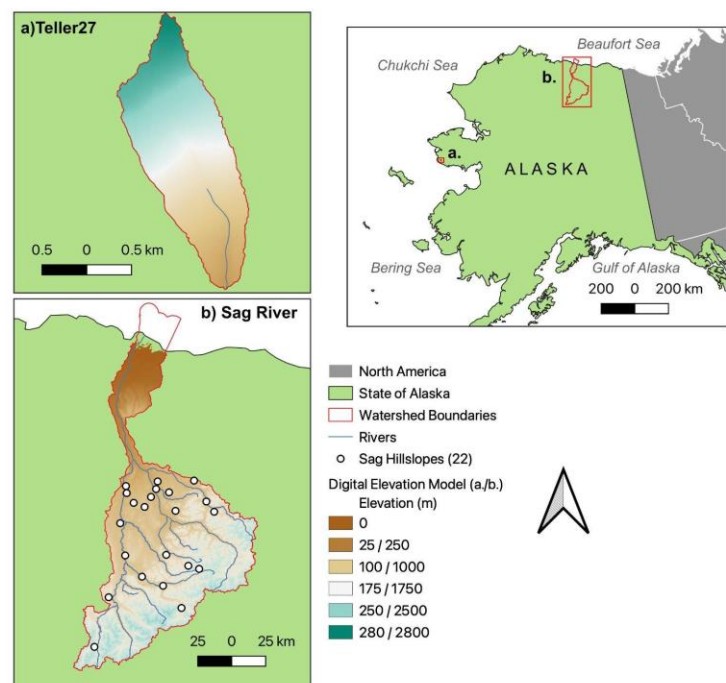

**Figure 1: Map of the a) Teller27 and the b) Sag River sites, and inset map of Alaska with sites indicated with red boxes. Detailed site maps a) and b) include watershed boundaries (red), rivers (blue), Sag River hillslopes (white dots, 22), and the Digital Elevation Models (DEMs) for each site. The 2.5 km2 Teller27 site, being much smaller and with much more limited elevational range than the Sag River site, is 10 times smaller than the Sag River basin elevational range (see legend). Sag hillslopes (22) are not found on the lower elevation North Slope coastal plain, below 250 m.**

### 3.2 ELM runoff parameterization evaluation

We constructed variable-width hillslope models for each of the 22 hillslope sites to represent both vertical and lateral dynamics, and simulated the fully coupled cryo-hydrological processes using ATS. To facilitate comparison with ELM, we

post-processed the ATS simulation outputs by averaging the surface and subsurface variables across ATS's multiple horizontal units (columns), preserving the number of vertical layers (rows, Figure 2). This produced a 1D multi-layer (rows), single column profile that mimics ELM's 15-layer subgrid-scale structure. From this 1D multi-layer representation, key



model variables such as maximum seasonally thawed depth of the upper soil layers above the permafrost (referred to as active layer thickness, ALT), surface water content, water table depth, soil moisture and ice content were extracted and used

as inputs to ELM's runoff parameterization equations (Eqs. (1)–(4)) to compute surface and subsurface runoff components. The resulting total runoff from ELM was then compared against the cumulative runoff stimulated directly by ATS. Figure 2 illustrates the overall workflow, including the extraction and processing of ATS's variables, the calculation of ELM runoff components, and the comparison with ATS's discharge accumulation at the outlet, as well as the averaging of ATS horizontal units to a 1D multi-layer representation. Layers can be of variable depth, as indicated in Figure 2.

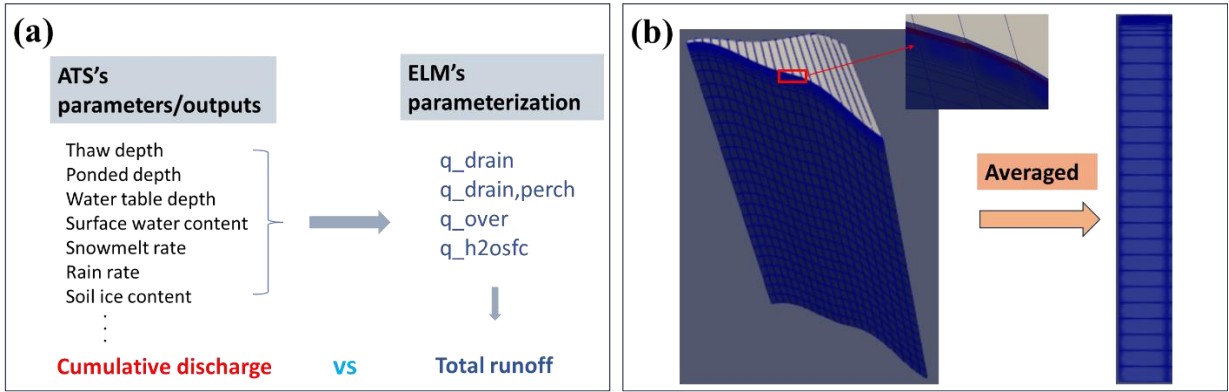


**Figure 2: Map (a) Workflow for calculating ELM's total runoff, highlighting key variables, runoff components, and computational steps involved. (b) Averaging of ATS's simulation results, illustrating the transformation from high-resolution pseudo-2D ATS outputs to simplified 1D ELM column representations.**

To obtain the optimized runoff coefficients, we formulated objective functions that minimize the difference between the ATS-simulated runoff and a weighted sum of ELM's runoff components. The optimization was carried out using a constrained numerical minimization algorithm, with non-negativity bounds imposed on all coefficients. This procedure was conducted across multiple hillslope sites and under both annual and seasonal (warm–cold) conditions, resulting in site-independent adjusted coefficients for each case. To assess the performance of the optimized runoff coefficient and quantify

the differences between them, three statistical metrics were calculated: root mean square error (RMSE), mean absolute error (MAE), Nash–Sutcliffe efficiency coefficient (NSE). They are defined as follows:

$$RMSE(c_1, c_2, c_3, c_4) = \sqrt{\frac{1}{n}\sum_{i=1}^{n}\left(y_i - \hat{y}_i\right)^2} \tag{11}$$

$$MAE(c_1, c_2, c_3, c_4) = \frac{1}{n}\sum_{i=1}^{n}\left|y_i - \hat{y}_i\right| \tag{12}$$





$$NSE(c_1, c_2, c_3, c_4) = 1 - \frac{\sum_{i=1}^{n} (y_i - \hat{y}_i)^2}{\sum_{i=1}^{n} (y_i - \overline{y}_i)^2} \tag{13}$$

where $y_i$ and $\overline{y}_i$ represent the ATS simulated value and its average value, respectively;

$\hat{y}_i = c_1 q_{drain} + c_2 q_{drain,perch} + c_3 q_{over} + c_4 q_{h2osfc}$ is the calculated (or optimized) value based on ELM's equations. The

coefficients $c_1$, $c_2$, $c_3$ and $c_4$ are weighting factors used to evaluate the relative contribution of each component. In the non-

adjusted case, all four coefficients are set to 1.0, directly summing all runoff components. In the adjusted case, the

coefficients are optimized through regression against the ATS-simulated runoff to best match the observed behavior. $n$ is

the number of annual or seasonal cumulative runoff values used in the comparison.

## 4. Results

### 4.1 Evaluation of the runoff schemes in ELM

ELM runoff results were evaluated in comparison with ATS's drainage at the outlet. In the plots below, we compared the

annual and seasonal total cumulative runoff using both adjusted and non-adjusted coefficients $c_i$ , under a variety of

conditions across different hillslope sites (e.g., forcings, slope, ALT, etc.). The three columns in Figure 3 correspond to

different ELM parameterization cases: (1) unadjusted coefficients, (2) annually optimized coefficients, and (3) seasonally

(warm–cold) optimized coefficients. The three rows represent different color-coded variables: slope (top), ALT (middle),

and total precipitation (bottom).





**Figure 3. Comparison of cumulative total runoff between ATS and ELM models under various hillslope conditions. The different colors represent slope (a-c), active layer thickness (d-f), and total precipitation (g-i). Columns (from left to right) correspond to different coefficient settings in ELM: unadjusted (a, d, g), annually adjusted (b, e, h), and seasonally adjusted (c, f, i). The results are displayed for different runoff parameterization coefficients (as indicated on the y-axis) in ELM and compared against ATS-simulated runoff.**

It can be seen that ELM's annual total cumulative runoff is generally smaller than ATS's runoff when using unadjusted runoff coefficients, with a NSE of 0.30 (Figure 3a). However, when adjusted coefficients are applied, the two models show



much better agreement, achieving an NSE of 0.68 when adjusted based on annual discharge (Figure 3b) and 0.83 when optimized using separate adjustments for the warm and cold periods (Figure 3c). The seasonal total cumulative runoff is
calculated for the warm season (May to August) and cold season (September to April) within the calendar year. This improved alignment in the seasonal total cumulative runoff is also reflected in performance metrics, with the RMSE, MAE, and NSE increasing by 39 %, 41 %, and 22 %, respectively, over the total runoff without adjusted coefficients, respectively.

The total runoff during the cold season is normally lower than that of the warm season. This is primarily because when the ground is fully frozen, overland flow ($q_{over}$) becomes the only dominant component of total runoff, which occurs due to
excess meltwater from snow after limited vertical infiltration. As shown by the diamond symbols in Figures 3c, 3f, and 3i, ELM's results are generally well-aligned with ATS's drainage during cold seasons, though some values are slightly lower. In the cold season, no active layer exists, whereas it does exist during the warm season. Cold-season runoff on steeper slopes tends to be lower, likely due to limited infiltration in frozen soils, which reduces subsurface flow pathways. In contrast, during the warm season, total runoff generally increases with slope, as shown in Figure 3c, likely due to enhanced overland
flow and faster hydrological response. Despite this trend, the runoff alignment in the seasonal total runoff (Figure 3c) remains better compared to the total annual (Figure 3b). This indicates that runoff differences between the two models cannot be fully explained by variations in slope, highlighting the influence of additional factors such as model parameterizations and physical processes.

By carefully examining the results with the adjusted coefficients in Figure 3b, we observed that ELM performs well in
representing lower slopes compared to the ATS benchmark, with a few exceptions. On steeper slopes (> 8 degrees), ELM predicts lower runoff values than ATS, with differences reaching up to 150 kg·m$^{-2}$. Interestingly, ELM also underpredicts runoff on shallower slopes (< 2 degrees), suggesting that runoff differences between the two models are influenced by more than just slope. This variation could arise from differences in how the two models handle topographic gradients, basin size, or spatial heterogeneity in climate forcings. Figure 3e highlights that ELM performs better with deeper ALT, likely because
these active layers experience less freezing, allowing for increased water storage in the deeper thawed zones and reduced lateral drainage. Figure 3h reveals a strong relationship between total cumulative runoff and precipitation, and ELM generally captures lower precipitation events more accurately, leading to correspondingly lower total runoff. As expected, Figure 3i reveals a clear trend between warm-season total runoff and precipitation magnitude. However, ELM tends to underpredict runoff under higher precipitation conditions. This underscores the critical role of climate forcing data in
shaping model performance and highlights ELM's sensitivity to precipitation variability.

To better understand these findings, it may be important to consider spatial variations in terrain landforms (e.g., concave vs. convex) across the modeled hillslopes. In addition, these watersheds span a broad latitudinal range (approximately 68° to 70°N), which likely introduces considerable climatic and topographic variability. For example, steeper slopes can promote rapid surface runoff, while gentler slopes with deeper thawed layers may enhance water retention and reduce lateral
drainage.





This improved alignment between ELM and ATS's annual total cumulative runoff is also evident in time series comparisons of total runoff from a selected hillslope, shown in Figure 4, both with and without adjusted coefficients Figure 4a illustrates the results using the unadjusted coefficients, where ELM underestimates runoff peaks and exhibits a relatively weak correlation with ATS discharge. The RMSE, MAE, and NSE values (0.55, 0.22, and 0.48, respectively) indicate moderate

agreement but suggest room for improvement, particularly in capturing peak runoff events. Figure 4b shows the impact of applying annually adjusted coefficients, which improves model performance, as reflected by reduced RMSE (0.51) and MAE (0.19), along with a higher NSE (0.56). The adjusted coefficients result in better agreement during high-runoff periods, though some discrepancies remain in capturing the full variability of runoff responses. Figure 4c presents the results with seasonally adjusted coefficients, which yield the best overall performance among the three cases. The RMSE and MAE

decrease further to 0.46 and 0.18, respectively, while the NSE increases to 0.64, demonstrating a stronger correlation between ELM and ATS runoff. The seasonal adjustment appears to enhance the model's ability to capture the timing and magnitude of runoff peaks, particularly during snowmelt periods. However, some deviations remain, likely due to limitations in parameterizing snowmelt-driven surface flow and subsurface hydrological interactions in ELM.

These results highlight the importance of refining runoff coefficients and incorporating seasonal variations to improve the

predictive capability of ELM, particularly in Arctic environments where snowmelt dynamics play a dominant role in runoff generation. In general, better alignment is observed when ATS's runoff remains below a certain threshold, such as 150 kg·m$^{-2}$. For larger runoff, ELM tends to underpredict total runoff, indicating limitations in its ability to represent higher runoff scenarios accurately.




**Figure 4. Time series comparison of total runoff between ATS and ELM for a typical hillslope with an average slope of 6.6°, a watershed area of 2.43 km2, and the ALT ranging from 0.43 to 0.59 meters. Results are shown with (a) unadjusted coefficients, (b) annually-adjusted coefficients, and (c) seasonally-adjusted coefficients.**

### 4.2 Teller27 watershed evaluation

The improved ELM runoff coefficients were evaluated using observed streamflow data from the Teller27 watershed, spanning 2016–2019 (Busey et al., 2019). Results in Figure 5 indicate that the adjusted coefficients significantly enhance the agreement between ELM-simulated runoff and observations. The unadjusted coefficients consistently underpredict runoff, with the largest discrepancy observed in 2016 (ELM: 374.66 kg·m$^{-2}$, observed: 648.25 kg·m$^{-2}$). Both annually and seasonally adjusted coefficients reduce this gap, with the seasonally adjusted coefficients providing better performance, particularly in years with greater interannual variability in precipitation and thaw depth.





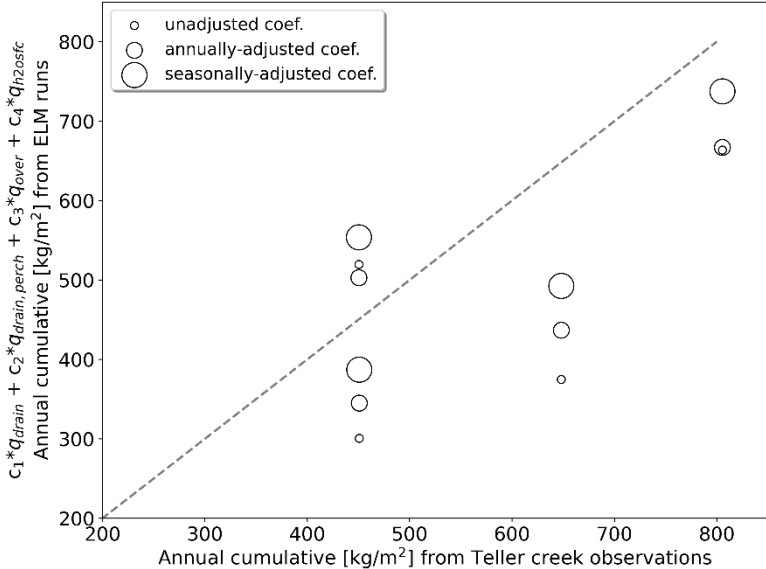

**Figure 5. Comparison of cumulative annual total runoff between ELM-simulated and observed data at the Teller27 watershed, Alaska, from 2016 to 2019. Symbol sizes represent ELM results using unadjusted coefficients, annually adjusted coefficients, and seasonally adjusted coefficients.**

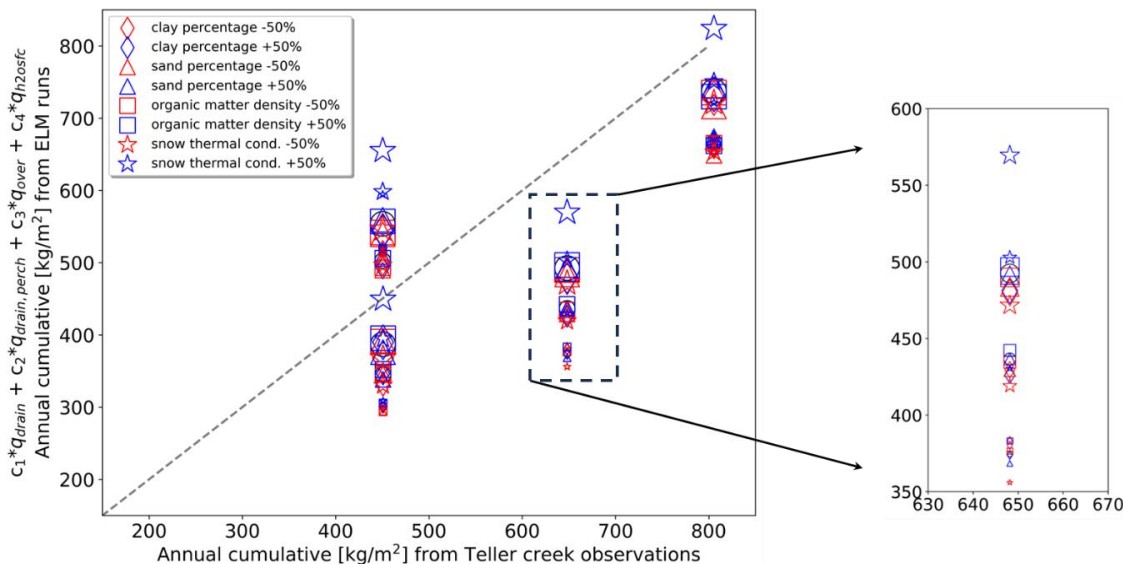

**Figure 6. Differences in annual cumulative total runoff from ELM simulations at the Teller27 watershed, Alaska. Different symbols indicate scenarios where one parameter is reduced (red) or increased (blue) by 50 % from its default average values in ELM's soil physical property and snow thermal conductivity, while symbol sizes represent results with unadjusted coefficients (small), annually adjusted coefficients (medium), and seasonally adjusted coefficients (large), as in Figure 5.**





Figure 6 presents the outcomes of the parameter sensitivity experiments, illustrating how changes in land surface properties influence simulated total runoff. It can be seen that total runoff is relatively insensitive to variations in surface soil properties such as clay, sand, and organic matter content, suggesting that soil texture and composition play a secondary role in controlling runoff dynamics within the model. Simulated runoff totals under these parameter perturbations remained within a relatively narrow envelope, regardless of whether baseline or optimized runoff coefficients were applied. This occurs because runoff is most due to overland flow during the spring snowmelt, when the ground is still frozen. These differences were notably smaller than the variations observed across different years or watersheds. The broader implications of these findings are further discussed in Section 5.2.

Figure 7 shows a temporal comparison of weekly and monthly total runoff, while Table 1 quantifies performance using RMSE, MAE, and NSE metrics. The adjusted coefficients yield lower RMSE and MAE values and higher NSE scores than the unadjusted coefficients, confirming their effectiveness, despite being derived from a study site some distance from the Teller27 watershed. These results clearly demonstrate that ELM's subgrid-scale runoff scheme performs well at the monthly timescale, but struggles to capture sub-monthly (e.g., weekly) runoff variability. Although ELM's total runoff is of the same order of magnitude as the observed discharge at Teller creek, discrepancies arise during specific events. Notably, spikes in runoff occur during rapid snowmelt in early spring due to ponded surface water, and the baseflow during recession periods are poorly simulated.

The weekly average results (Figure 7a) reveal significant discrepancies in runoff peaks with unadjusted coefficients, while those biases are reduced by the simulations with adjusted coefficients. This may be due to large peak flow simulated during snowmelt in ELM that is not represented in the observations, thus flows throughout the rest of the summer are too low in the simulations. We believe that this response may be occurring due to a frozen active layer that leads to fast runoff that is not observed in the hydrological records. While we observed this overestimate of flow peaks, the seasonally adjusted coefficients still perform better in capturing both the timing and magnitude of runoff peaks, perhaps because the seasonal coefficients adjust for some attenuation of runoff. The monthly results (Figure 7b) show improved agreement with observed data during high runoff periods, particularly those driven by snowmelt and seasonal precipitation. This is likely because the monthly results accumulate flow responses over longer periods of time, accounting for and spreading out this large peak flow runoff response that was incorrectly simulated at the weekly time scale. For example, at the monthly scale, NSE improves markedly from 0.07 with unadjusted coefficients to 0.58 and 0.60 with annually and seasonally adjusted coefficients, respectively. These results highlight the efficacy of the adjusted coefficients and the critical role of incorporating seasonal variability into runoff parameterization for improving ELM's performance.



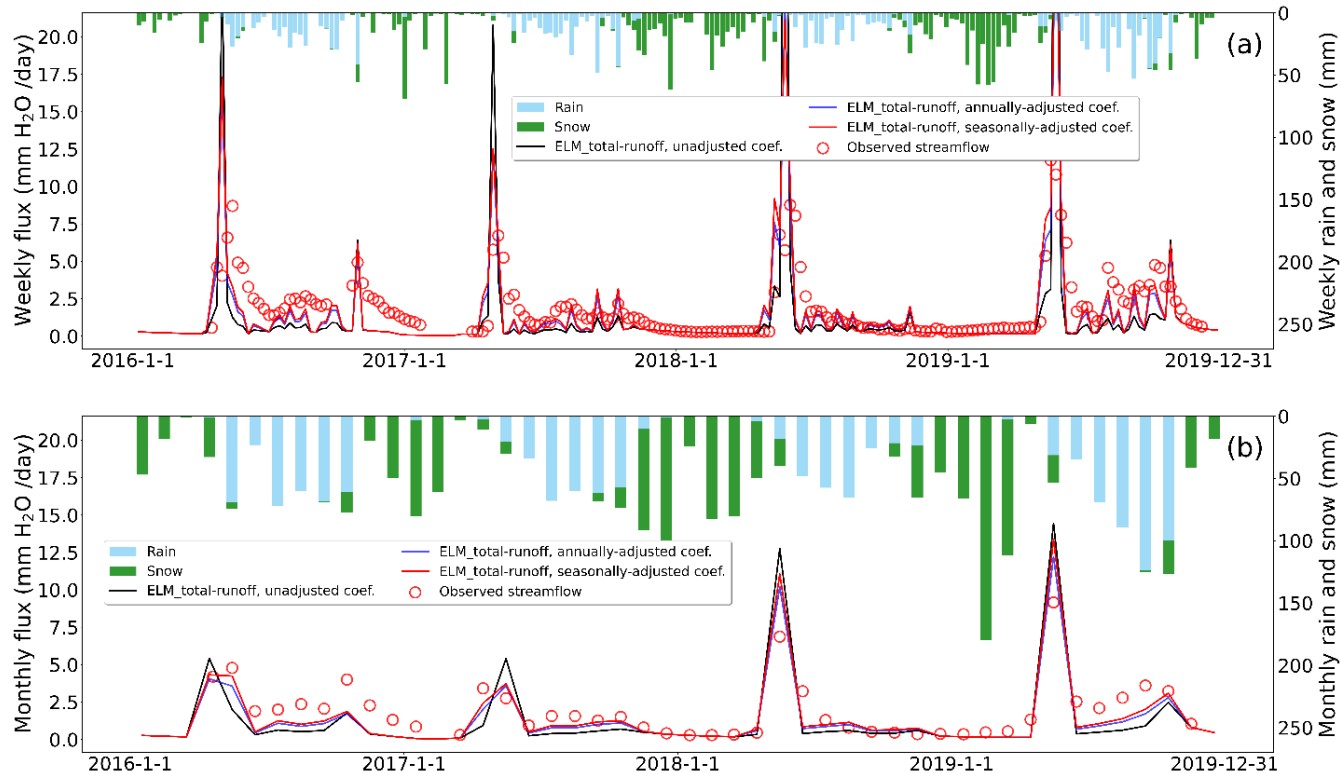

**Figure 7. Comparison of ELM-simulated (a) weekly and (b) monthly total runoff with observed data at the Teller27 watershed, Alaska. Simulated results are displayed using different runoff parameterization coefficients in ELM, represented by distinct colored lines. Rain and snow precipitation are shown as bar plots using the top X-axis (time) and left Y-axis (mm) in both subplots.**

**Table 1. Performance metrics (RMSE, MAE, and NSE) for ELM-simulated total runoff compared to observed streamflow across all temporal scales (daily, weekly, and monthly) at the Teller27 watershed, Alaska.**

| Temporal average of total runoff | Adjustment of ELM runoff coefficients | RMSE | MAE | NSE |
|---|---|---|---|---|
| | unadjusted | 12.66 | 1.96 | -32.16 |
| Daily | annually-adjusted | 5.80 | 1.62 | -5.95 |
| | seasonally-adjusted | 5.55 | 1.66 | -5.37 |
| | unadjusted | 4.91 | 1.73 | -4.80 |
| Weekly | annually-adjusted | 2.61 | 1.22 | -0.64 |
| | seasonally-adjusted | 2.49 | 1.20 | -0.48 |
| | unadjusted | 1.76 | 1.20 | 0.07 |
| Monthly | annually-adjusted | 1.18 | 0.84 | 0.58 |
| | seasonally-adjusted | 1.16 | 0.81 | 0.60 |





## 5. Discussion

Intercomparison studies to consider differences and similarities between a land surface scheme to a physically-based model
are useful because they allow for quantitative evaluation of each model and its performance for different components of the
hydrologic cycle. The technique applied here was to extract the land surface parameterization schemes and evaluate them
within a fine-scale, physics model, to consider the total cumulative runoff from the land surface scheme and compare it with
the physics model estimates. This novel approach allowed us to focus on the formulations over other interacting effects that
may have potentially obscured the results. We were then able to make an estimate of runoff coefficients that could be applied
to improve the match between the physics-model and the land surface model. These coefficients, when applied to a small
watershed located in sub-arctic Alaska, improved the land surface model significantly, suggesting that this approach may
allow for fine-tuning of runoff within similar systems. Overall, further research is needed to determine how flexible these
coefficients are; however, this work was foundational in that it provided a novel approach to both model intercomparison as
well as for model improvement.

A sensitivity analysis conducted as part of this study identified that soil properties may exert less influence on the runoff
dynamics in the land surface model than snow accumulation and melt processes, as well as subsurface thermal hydrology.
This finding links this component of our research to investigations into snow processes and model improvement in these
same systems (e.g., Bennett et al. 2022; Clark et al. 2015). It is clear that to improve runoff processes within permafrost
dominant systems, a holistic approach to understanding the system is required to determine the key processes and model
components that require adjustment. This study provided an initial enquiry into this process that developed the foundation
for our work towards improving the earth system model for these Arctic systems.

### 5.1 Differences in simulated runoff from representative hillslopes

The apparent mismatch between ELM and ATS runoff results from 22 representative hillslopes in the Sag River basin
warrants a critical examination. However, such evaluations must consider the inherent differences between the two models.
Several factors could contribute to these differences. First, a key distinction between ATS and ELM is their treatment of
runoff generation processes. ATS implicitly simulates variably saturated flow, lateral subsurface flow and transport, and
dynamic freeze-thaw cycles in a high-resolution 3D domain (Painter et al., 2016; Coon et al., 2020). In contrast, ELM
utilizes parameterized equations to represent subgrid-scale heterogeneities, which is not expected to fully capture the
complexity of permafrost hydrology (Bisht et al., 2018; Xu et al., 2024). This averaging process that reduces a 3D system
into a pseudo-3D system in ATS can represents a loss of spatial variability in hydrological processes, particularly in
heterogeneous permafrost regions where local-scale topography and subsurface heterogeneity play a critical role in runoff
generation (Zhao & Li, 2015; Abolt et al., 2024). This simplification can also lead to discrepancies in how surface and
subsurface runoff are partitioned (Liao et al., 2024), particularly in regions with high spatial variability in soil moisture, ice



content, and thaw depth. For instance, ELM may under- or over-predict runoff during high precipitation events (see Figure
5) compared with the spatially resolved, physics-based implementation in ATS.

Secondly, although ATS is treated as the benchmark in this study, it is not without uncertainties. The accuracy of ATS
simulations depends on input data such as soil properties, meteorological forcings, and initial conditions, all of which contain
inherent uncertainties (Harp et al. 2016; Jafarov et al., 2018; Zhang et al., 2023; Huang et al., 2024). In permafrost
landscapes, soil heterogeneity is particularly difficult to characterize, and small variations in soil thermal and hydrological
properties can lead to substantial differences in runoff predictions (Decharme et al., 2013; Vereecken et al., 2022).

Additionally, the accuracy of ELM's runoff predictions is highly dependent on the parameterization of surface hydrological
processes. Although optimized coefficients have been implemented to improve agreement with ATS, these parameterizations
may still inadequately capture the nonlinear interactions between infiltration, permafrost thaw, and lateral flow (Swenson et
al., 2012; Liao et al., 2024). This issue is particularly relevant in ice-rich permafrost terrains, where abrupt changes in thaw
depth and active layer dynamics can lead to nonlinear responses in runoff generation (e.g., Hinzman et al., 2022). The lack of
explicit lateral flow representation in ELM further limits its ability to capture runoff redistribution processes that are well-
resolved in ATS simulations.

Another factor that may contribute to the mismatch is the difference in how the two models resolve seasonal freeze/thaw
processes, especially under varying precipitation and thawing conditions. Runoff generation in permafrost regions is highly
sensitive to seasonal thawing and freezing dynamics, as well as precipitation regimes (e.g., Zhang et al., 2010; Guo et al.,
2025). In particular, mismatches may arise during transitional periods such as spring snowmelt, when small differences in
temperature and soil conditions can lead to substantial variations in runoff production.

Understanding these potential differences is crucial for interpreting the model responses and guiding future ELM model
improvements. Addressing the uncertainties in ATS and refining the transformation of parameterization schemes between
models could reduce these mismatches. Similarly, enhancing ELM's parameterization by incorporating insights from ATS
simulations, such as better representation of lateral flow and freeze-thaw processes, could lead to improved alignment with
ATS and more accurate predictions in Arctic environments under changing climate.

**5.2 Sensitivity and uncertainties of model performance to land surface parameters in ELM**

The sensitivity analysis assessed the impact of various surface and subsurface parameters, including soil properties (clay,
sand, and organic content) and snow thermal conductivity, on the simulated total runoff. Understanding the sensitivity of
runoff simulations to these parameters is crucial for improving hydrological predictions in permafrost regions, where land
surface processes interact with freeze-thaw dynamics in complex ways (Bisht et al., 2018; Walvoord & Kurylyk, 2016).

The limited sensitivity of total runoff to soil property variations observed in our ELM simulations raises important
implications for land surface model development. Previous studies have shown that soil texture can strongly affect
bidirectional water exchange between groundwater and the soil during freeze–thaw transitions (e.g., Xie et al., 2021; Huang



& Rudolph, 2023; Yang et al., 2025). However, our findings suggest that these processes may have less influence on annual runoff generation in permafrost regions. This discrepancy may be explained by the dominant role of snowmelt dynamics and shallow subsurface hydrology in controlling surface runoff. In permafrost-dominated landscapes, runoff generation is often driven by the timing and intensity of snowmelt, seasonal freeze–thaw cycles, and the spatial distribution of near-surface

permafrost. These factors likely outweigh the direct effects of variations in soil grain size or organic matter content (e.g., Swenson et al., 2012). The presence of an impermeable permafrost layer beneath the active layer restricts deep infiltration and causes excess water to remain near the surface, limiting the direct impact of soil properties on runoff partitioning. Similar findings have been reported in other permafrost hydrology studies, where hydraulic conductivity and soil texture exert minimal influence on runoff formation compared to freeze-thaw dynamics and snowmelt timing (e.g., Zhang et al.,

1999; Walvoord & Kurylyk, 2016). However, further investigation is needed to evaluate whether subsurface water redistribution, active layer depth variability, and lateral flow dynamics could play a more significant role in influencing ELM's runoff performance.

In contrast to soil parameters, snow thermal conductivity exhibits a strong influence on simulated runoff, demonstrating its critical role in shaping hydrological responses in Arctic environments. An increase in snow thermal conductivity enhances

heat transfer within the snowpack, leading to earlier and more rapid snowmelt. This, in turn, alters the seasonal timing of water availability and increases runoff magnitudes during peak melt periods (Musselman et al., 2017). Higher thermal conductivity results in faster warming of the snowpack, reducing the buffering effect of snow insulation and exposing the underlying soil to greater temperature fluctuations. This phenomenon has been observed in field studies, where changes in snow properties significantly impact the timing and magnitude of spring runoff (Würzer et al., 2016; Liljedahl et al., 2016).

The results in Figure 6 suggest that accurate representation of snow properties is essential for improving runoff predictions in permafrost landscapes. Over- or under-estimating snow thermal conductivity could lead to systematic biases in modeled runoff timing, potentially affecting the accuracy of hydrological assessments in Arctic watersheds.

Collectively, these findings reinforce the critical role of snowmelt-driven hydrological processes in shaping runoff dynamics in permafrost landscapes and illustrate key sensitivities within ELM's runoff parameterization. The results suggest that

model performance is particularly influenced by representations of snow accumulation and melt processes, as well as subsurface thermal hydrology. In particular, sensitivity to snow density, thermal conductivity, and freeze–thaw transitions points to the value of incorporating physically based formulations that capture snowpack variability (e.g., Lackner et al., 2022; Tao et al., 2024) and lateral subsurface flow (e.g., Swenson et al., 2012; Liao et al., 2024). These process-level influences appear to exert a stronger control on runoff behavior than surface soil properties alone, underscoring their

importance in cold-region hydrology and land surface modeling.



### 5.3 Implications for improving runoff parameterization coefficients in land surface models

Hydrological runoff-related parameters in land models are calibrated against the observed streamflow data (e.g., Niu et al., 2007; Li et al., 2013), which is often limited or unavailable in remote permafrost regions. This study introduces a novel evaluation framework, which shifts the traditional paradigm of directly comparing coarse-scale land surface models to fine-scale physics-based models, by deriving optimized runoff coefficients by leveraging high-fidelity simulations from the integrated surface/subsurface hydrological simulators such as the ATS. These optimized coefficients are then incorporated into the land surface model, allowing for physics-informed improvements without direct site-based calibration. This approach offers a process-oriented alternative to conventional calibration, providing an avenue for improving parameterizations in data-scarce regions.

A key implication of this framework is its potential transferability across diverse Arctic watersheds. In this study, coefficients derived from the Sag River hillslopes were applied to the Teller27 watershed without additional tuning, resulting in significantly improved runoff performance at monthly and seasonal scales. One possible reason for this is that while these two systems are located in very different environments and some distance from each other, they are similar in that both exhibit moderately graded slopes and elevations, vegetation of grasses and shrubs, and both have some degree of permafrost (albeit discontinuous permafrost in Teller27). This demonstration suggests that physically guided coefficients obtained through fine-scale process-resolving models may be generalized to other Arctic catchments with similar characteristics, offering a possible strategy for parameter refinement in Earth system land models. Such transferability is especially valuable for land model intercomparison projects that seek robust parameterizations applicable across diverse permafrost regions (e.g., Clark et al., 2015; Lawrence et al., 2019; Fan et al., 2019). However, more research is required to determine the extent of this. Future applications of this approach at permafrost sites across the pan-Arctic, which is part of the next phase of this project, could further enhance the robustness and generalizability of this framework.

### 6. Limitations and future work

Based on our analysis and discussion, we acknowledge several limitations that may be further improved in future studies:

1. Simplified watershed representation. The hillslopes used for ATS simulations in the Sag River basin are pseudo-2D variable-width simplifications of the 3D landscapes, which do not fully capture the heterogeneity of real Arctic landscapes, such as ice-wedge polygons, thermokarst features, and microtopographic variations. Future studies should incorporate many more diverse landscapes (up to hundreds of hillslope models) and ensure identical topographic representations across both models.
2. Transferability evaluation. The optimized runoff coefficients were derived from ATS simulations of Sag River hillslopes and then directly applied to the Teller creek watershed without site-specific adjustments. This transferability was only evaluated at a single site (Teller27), further evaluation across diverse Arctic watersheds with longtime streamflow measurements is needed to build broader confidence in the generalizability of the approach.





3. Limited consideration of lateral flow and subsurface heterogeneity. This study primarily focused on ELM's
   subgrid-scale runoff generation, neglecting lateral water movement and groundwater interactions. In Arctic
   environments, lateral subsurface flow can play a crucial role in redistributing water across permafrost
   landscapes, affecting both surface runoff and baseflow dynamics in different land units, which will be
   evaluated in the next phase of the project.

4. Limited assessment of meteorological forcing biases. While the parameterization in this study is based on state
   variables derived from ATS simulations, rather than directly on precipitation or other forcings, model
   performance in applications such as the Teller creek test can still be sensitive to uncertainties in meteorological
   inputs. In high-latitude regions, sparse station coverage and undercatch issues can introduce substantial
   uncertainty in precipitation, temperature, and radiation datasets. Future work should assess how these
   uncertainties propagate through the model and influence runoff simulations under different forcing scenarios.

## 7. Conclusions

We evaluated a land surface model's runoff parameterization using detailed fine-scale physics-based simulations of 22 hillslopes in the Sag River basin and identified empirical adjustments that improve the runoff parameterization. Seasonal optimization of these coefficients improved the model's ability to capture hydrological variability at monthly scales, particularly in snowmelt-driven runoff processes. The adjusted parameterization improved ELM simulated runoff from the Teller27 watershed. That demonstration of transferability of the adjusted parameterization is encouraging, but needs further study across diverse Arctic catchments. Sensitivity analysis revealed that runoff in ELM is largely insensitive to soil properties but highly sensitive to snow thermal conductivity, underscoring the importance of accurate snow process representation in permafrost regions. These findings demonstrate the value of spatially resolved fine-scale simulators from physics-based models as benchmarks for refining land surface models and highlight the need for process-specific parameterization improvements in hydrological runoff schemes of land surface models.

Despite these advancements, challenges remain in capturing subsurface hydrological processes, including lateral flow, permafrost thaw dynamics, and active layer variability, which are critical for Arctic runoff simulations. Future improvements should focus on incorporating water redistribution within an ELM gridcell due to lateral flow, refining subgrid-scale hydrological parameterizations, and evaluating model updates across diverse Arctic catchments. By addressing these gaps, land surface models could achieve more accurate runoff predictions, ultimately enhancing their utility for climate impact assessments and water resource management in permafrost regions.

*Code and data availability.* All data sets used in this work are archived at the Environmental System Science Data Infrastructure for a Virtual Ecosystem (ESS-DIVE). ELM and ATS data sets will be archived here: https://doi.org/10.15485/2550570. Data sets will become live once the paper is accepted. ERA5 forcing data to run the ELM model can be downloaded from the ECMWF Climate Data Store: https://cds.climate.copernicus.eu/datasets/reanalysis-era5-single-levels?tab=overview. ATS forcings data can be retrieved from Daymet version 4 dataset (Thornton et al., 2020).



Streamflow data was derived from Busey et al. (2019): Surface Water: Stage, Temperature and Discharge, Teller Road Mile Marker 27, Seward Peninsula, Alaska, beginning 2016. Next-Generation Ecosystem Experiments (NGEE) Arctic, ESS-DIVE repository. Dataset. https://doi.org/10.5440/1618330. Data sets are in the transfer process between the NGEE Arctic data portal and the ESS DIVE repository, and will be updated as soon as possible. The description and codes of E3SM v3.0 (including ELM v3.0) are publicly available at https://www.osti.gov/doecode/biblio/123310 (E3SM Project, 2024) and https://github.com/E3SM-Project/E3SM/releases/tag/v3.0.0 (released: 4 March 2024), respectively.

*Author contributions.* The study was conceptualized and designed by YZ, SP, XH and KB. XH has done the data analysis, visualization, and writing the original draft. All authors contributed to editing the manuscript.

*Competing interests.* The authors declare no competing interests.

*Acknowledgments.* The authors gratefully acknowledge Ian Shirley and Baptiste Dafflon for their support with data curation at the Teller Creek watershed. We also thank Peter Thornton for his technical assistance with the implementation and interpretation of the E3SM-ELM model. This work is financially supported by the Next Generation Ecosystem Experiment (NGEE) Arctic project from the Office of Biological and Environmental Research in the U.S. Department of Energy's Office of Science.

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
