# Peer review of "Runoff Evaluation in an Earth System Land Model for Permafrost Regions"

_EGUsphere, 2025_

## Community Comment (CC2)

**Code and data availability**

All data sets used in this work are archived at the Environmental System Science Data Infrastructure for a Virtual Ecosystem (ESS-DIVE). ERA5 forcing data to run the ELM model can be downloaded from the ECMWF Climate Data Store: https://cds.climate.copernicus.eu/datasets/reanalysis-era5-single-levels?tab=overview. ATS forcings data can be retrieved from Daymet version 4 dataset (Thornton et al., 2020). ELM and ATS data sets are archived here: https://doi.org/10.15485/2550570. Streamflow data was derived from Busey et al. (2019): Surface Water: Stage, Temperature and Discharge, Teller Road Mile Marker 27, Seward Peninsula, Alaska, beginning 2016. Next-Generation Ecosystem Experiments (NGEE) Arctic, ESSDIVE repository. Dataset. https://doi.org/10.5440/1618330. The description and codes of E3SM v3.0 (including ELM v3.0) are publicly available at https://www.osti.gov/doecode/biblio/123310 (E3SM Project, 2024).

[Figure]

Huang X ; Gao B ; Demir C ; Fiorella R ; Painter S ; Bennett K (2025): Data Files for Runoff Evaluation in an Earth System Land Model for Permafrost Regions. Next-Generation Ecosystem Experiments (NGEE) Arctic, ESS-DIVE repository. Dataset. doi:10.15485/2550570 accessed via https://data.ess-dive.lbl.gov/datasets/doi:10.15485/2550570 on 2025-06-27

Busey B ; Wales N ; Newman B ; Wilson C ; Bolton B (2019): Surface Water: Stage, Temperature and Discharge, Teller Road Mile Marker 27, Seward Peninsula, Alaska, 2016-2021. Next-Generation Ecosystem Experiments (NGEE) Arctic, ESS-DIVE repository. Dataset. https://doi.org/10.5440/1618330.

DOE Code Repository Software Policy

https://www.osti.gov/doecode/policy

---

## Community Comment (CC5)

**GMD Review Response – Reviewer #1**

The manuscript titled "Runoff Evaluation in an Earth System Land Model for Permafrost Regions" proposes a new framework using ATS simulations to optimize runoff coefficients, offering a more physically based alternative to traditional parameterizations in Earth system land models. Furthermore, the study explores the potential transferability of the optimized runoff coefficients across different Arctic watersheds. This work presents methodological innovations that are valuable for improving runoff simulation in permafrost-affected regions and holds implications for enhancing water resource management in high-latitude environments.

The manuscript is generally well-written, and contributes meaningfully to the field. However, there are several issues that should be addressed to improve the clarity and precision of the manuscript. I therefore recommend the manuscript a minor revision, by considering some comments and questions posed below.

Response: Thank you for taking the time to review our paper.

While the study proposes a broadly applicable framework, the current analysis is limited to only two watersheds in Alaska: the Sagavanirktok (Sag) River Basin and the Teller watershed. Given this limited spatial extent, the current title may overstate the geographic generalizability. A more precise title such as "Runoff evaluation of an Earth System land model in the permafrost region of Alaska" would better reflect the study's current scope.

Response: Thank you. We have adjusted the paper's title as follows: "Runoff Evaluation in an Earth System Land Model for Permafrost Regions in Alaska"

Please correct the unit formatting on Line 316: "km2" should be changed to "km²" (with superscript).

Response: Thank you. We have corrected this typographic error.

Figure 6 attempts to illustrate the effect of changes in soil physical properties and snow thermal conductivity on runoff simulations within the ELM model. However, the current visualization makes it difficult for readers to extract meaningful comparisons. Consider redesigning the figure to improve visual clarity. For example, using grouped bars or difference plots to highlight contrasts between scenarios.

Response: We agree, and we have revised Figure 6 significantly in response (see below). In the caption of Figure 6, we now highlight that the barplot is showing the modeled (ELM) values, and the y axis is common for both modeled (histogram) and the observed Teller data (dashed line).

We added a Supplement to the paper with each year for Figure 6 shown in a separate panel (see below).

[Figure]

Figure R1.1. Revised Figure 6 from the article.

The new Figure 6 caption is as follows: Variations in annual cumulative total runoff from ELM simulations at the Teller27 watershed, Alaska for the mean of all years (2016-2019). Different bars groupings along the x-axis illustrate scenarios where parameters are reduced or increased from their default average values in ELM's soil physical property and snow thermal conductivity. Bar colours represent results with unadjusted coefficients (teal), annually adjusted coefficients (green), and seasonally adjusted coefficients (grey). The mean annual cumulative runoff from Teller observations (mean all years 2016-2019) is given in the dashed line.

[Figure]

Figure R1.1. Revised Figure 6 from the article that will be included in a Supplement.

In Figure 7a, the legend partially obstructs key elements of the plot. Since subplots 7a and 7b share the same legend, a common external legend (e.g., placed to the right of the panel or beneath both subplots) would declutter the figures and enhance readability.

Response: Also agreed on this point. We have adjusted Figure 7 as you suggest.

[Figure]

Figure R1.3. Revised Figure 7 from the article with a common legend below the figure.

---

## Author Comment (AC1)

**GMD Review Response – Reviewer #2**

This study evaluates hydrological runoff in two permafrost-affected Arctic watersheds, Teller and Sag River, using two different models: the high-resolution, physics-based 3D permafrost hydrological model ATS and the coarse-resolution 1D Earth system model E3SM-ELM.

The authors simulate 22 representative watersheds in the Sag River basin with the ATS model. The runoff calculated by ATS is then used to calibrate linear parameters of the ELM model. These parameters are calibrated both annually and seasonally. The results show that by weighting four discharge components that contribute to total runoff in ELM, the modeled runoff aligns more closely with ATS results. The main conclusion is that the linear parameters (c1, c2, c3, c4) derived for Sag River can be applied to Teller, suggesting potential scalability to other sites.

**Response: Thank you for taking the time to review our paper.**

The transferability of these linear parameters across Arctic watersheds depends on which of the four discharge components dominate at different times of the year. Seasonal variability plays a key role: snow is a major factor during winter and spring, while rainfall becomes more important in summer. At Teller, a deeper active layer and talik development may enhance year-round subsurface hydrology. This suggests that parameter transferability may work reasonably well for colder Arctic regions but may be less applicable to sub-Arctic environments.

Response: Thank you for this comment. This also aligns with how we are thinking. We believe that this transferability of these parameters may work well in some systems with similar seasonal variability, snow processes, and permafrost features. The next step in our work is to test our new topographically-driven runoff module that was developed and input into ELM in the last year at multiple basins across the pan-Arctic. This effort will enable us to test these parameters along with that new module to determine how the two different approaches vary across different climates, basin sizes, vegetation, and permafrost conditions.

Instead of only focusing on parameter transferability, it would be valuable to compare this approach with the hillslope model approximation proposed by the CLM group (described in Swenson et al., 2019 and latest results presented by David Lawrence at AGU 2024). How similar or different is the method presented here compared to the CLM approach? This discussion would strengthen the study.

Response: As noted above, we are in the process of starting our evaluation of our new runoff module for ELM (<a href="https://ngee-arctic.ornl.gov/models/integrated-model-2">https://ngee-arctic.ornl.gov/models/integrated-model-2</a> and [<a href="https://github.com/E3SM-Project/E3SM/pull/6718">https://github.com/E3SM-Project/E3SM/pull/6718</a>]). We will evaluate this at several different model evaluation sites across the pan Arctic to determine how this improves runoff simulation. We will also test the parameterizations put forward in this work at those new sites. This paper did not apply that methodology, nor did we turn on any of the sub-grid variable modes of ELM for this work, as this work was a precursor to all of that effort that culminated in our improved runoff method for Arctic systems. We believe that once we have those new sites set up, our new runoff module implementation tested, and results at three new additional evaluation sites, we will be able to evaluate those methods against the CLM implementation. However, we have added a discussion of the CLM implementation from Swenson et al. 2019 in the paper on lines [lines]

915-937] and also some more discussion of our next steps on lines [line 951] of our revised manuscript. Please also see our next response as well.

Overall, this is an interesting and relevant study addressing the critical issue of scaling hydrological processes from local high-resolution models to Earth system models. The calibration and parameter transferability approach is promising but needs a stronger discussion of its limitations, particularly in regions with different seasonal dynamics. Adding detailed model setup descriptions, more figures illustrating profiles and seasonal variations, and a clearer comparison with alternative modeling approaches (e.g., CLM hillslope parameterization) will significantly strengthen the paper.

Response: Thank you for this comment. We have now added a reference to the limitations and the need for future work [lines 950-951] of the revised paper. We have also added more information on the model set up (Section 3.2) of the track changed version. We have also revised our figures as suggested [Figure 2, 3, and 4] to add more detail and clarity. We discuss the CLM hillslope parameterization as well.

Swenson, S. C., Clark, M., Fan, Y., Lawrence, D. M., & Perket, J.(2019). Representing intrahillslope lateral subsurface flow in the community land model. Journal of Advances in Modeling Earth Systems, 11, 4044–4065. https://doi.org/10.1029/2019MS001833

H13R-03 Improving terrestrial hydrologic process representation in Earth System Models: Accounting for slope, aspect, and lateral water transfer through representative hillslopes, AGU 2024

Schädel, C., Rogers, B.M., Lawrence, D.M. et al. Earth system models must include permafrost carbon processes. Nat. Clim. Chang. 14, 114–116 (2024). <a href="https://doi.org/10.1038/s41558-023-01909-9">https://doi.org/10.1038/s41558-023-01909-9</a>

Response: Thank you for providing these references. We have added them to our paper.

**Editorial & Content Suggestions**

L18: Replace physics-rich with physics-based or high-fidelity physics.

Response: We replaced the instances of physics-rich with physics-based.

L50: Use Schaedel et al., 2024 as a better reference.

Response: Thank you, we replaced Harp et al. 2016 with Schädel et al. 2024.

**Introduction:**

Conclude by emphasizing the scaling challenge: ATS operates at meter-scale resolution, whereas ELM works at  $\sim$ 150 km scale.

Discuss how this study attempts to bridge this scale gap, either through parameterization improvement in global-scale models or by showing how local-scale models inform global simulations.

The latter angle may be a safer framing.

Response: We have added text to the introduction discussing how this study attempts to bridge the scale gap that exists between ATS and ELM models.

We added to the first paragraph the following sentence:

"At the same time, Earth system models and land surface models are designed for pan Arctic scale simulations, creating a strong mismatch between the scale at which these processes occur and the models designed to represent them (Lique et al. 2016)."

We added to the last paragraph the following sentence:

"The method we detail in this work directly addresses the scale gap between local- to global-scale process representation in models, using intercomparison with local-scale ATS simulations and parameters updates in ELM to improve Arctic runoff processes."

**Model Description Section**

Provide more details on the model setups for ATS and ELM:

- Initial and boundary conditions
- Number of soil layers
- Differences in thermal, hydrological, and surface/subsurface properties

Include a figure comparing these configurations.

Response: To answer this question, in part, we have revised a component of the paper where previously we had included both the study area description and the modeling approach description into one section. We have now revised Section 3.1 to only describe the study areas and added a new Section 3.2 that describes the numerical design implemented in the study.

In this work, we never directly compare or simulate the ATS and ELM model in the same areas. Rather, we use ATS model results within the ELM model to optimize runoff coefficients and then apply these within a new watershed. Below, we describe the initial and boundary conditions, number of soil layers, and thermal, hydrological, and surface/subsurface properties of the ATS simulations and the ELM formulations.

**ATS Model:**

Our ATS modeling workflow follows three main steps. First, a column model with initial temperature above freezing is frozen from bottom up by setting a constant temperature -10 °C at the bottom face until a steady-state frozen soil column is formed. In this step, only the subsurface flow-energy system is included. The pressure and temperature profiles of the frozen column is used to initialize spinup run for several years for each hillslope model until a cyclically steady status. In this step, the surface, subsurface, and surface energy balance full physics system is used. Each cyclically steady hillslope is then used as the initial condition for its real transient runs. For each hillslope, the bottom temperature is constant at -10 °C. Closed boundary conditions are applied to all subsurface faces, and surface faces except the surface outlet. At the surface outlet, constant zero head is applied.

Three soil layers are designed in both 1D column and 2D hillslope models, including the top two organic soil layers (i.e., acrotelm, catotelm) and the bottom mineral layer. The thickness and soil properties of the soil layers of each column of a 2D hillslope model is related to and estimated by its corresponding land cover type according to the study by O'Connor et al. (2020). We have added details of the ATS set up to the paper to further describe these features on [lines 222-289]. The soil properties of each soil layer are listed in the table below. The thermal conductivity model is from Atchley, et al. (2015).

Table R2-1. Properties of soil layers (acrotelm, catotelm, and mineral) used in ATS Sag River simulations applied in both ELM and ATS.

| Soil layer                                                           | Acrotelm | Catotelm | Mineral  |
|----------------------------------------------------------------------|----------|----------|----------|
| Porosity                                                             | 0.88     | 0.8      | 0.457    |
| Permeability (m 2 )                                       | 1.29e-10 | 4.72e-12 | 1.82e-13 |
| van Genuchten α (Pa -1 )                                  | 7.78e-4  | 1.71e-4  | 6.94e-05 |
| van Genuchten n                                                      | 1.41     | 1.57     | 1.54     |
| Residual saturation                                                  | 0.08     | 0.08     | 0.04     |
| Thermal conductivity, saturated (W m -1 K -1 ) | 0.52     | 0.63     | 1.34     |
| Thermal conductivity, dry (W m -1 K -1 )       | 0.07     | 0.09     | 0.23     |

Figure R2.1. ATS Modeling on hillslope\_id=n3069, spatially and temporally averaged over the hillslope.

**ELM Model:**

Initial and boundary conditions are described for ELM in Section 2.1 and 3.2 of the paper. There are 10 soil layers and 5 bedrock layers. There is no hydrological flow occurring at the bottom of 10 soil layers, and there is no thermal flow at the bottom of the 5 bedrock layers.

Figure R2.2. ELM modeling for Teller27 under the default 0.5-degree ELM model configuration.

**Ground Conditions**

The WT was initialized at 8.8 m. Show the initial profiles of:

- Liquid pressure vs. depth
- Liquid and ice saturation vs. depth
- Temperature vs. depth

Response: In ELM, the subsurface is initialized with a soil temperature of 274 K and liquid water content of 0.15 m³/m³, together with an unsaturated storage of 4000 mm and a water table depth of ~8.8 m. These settings follow the CLM4.5 defaults, where all vegetated and glacier land units start from a large unsaturated store and moderately deep-water table. They are not site-specific, but neutral initial conditions that equilibrate toward realistic hydrologic states during spin-up. This is described in Section 2.2 of the paper and in the new Section 3.2 describing study design.

In ATS, we first performed a bottom-up freezing of the 1D variably saturated unfrozen soil column to produce an initial permafrost column, which was run to a steady-state under a bottom boundary temperature of -9.0 °C and a no flux at the top boundary (impermeable insulation).

Add figures for seasonal profiles of ground temperature and liquid/ice saturation for both Sag River and Teller.

Response: See above response and figures.

**Meteorological Data**

Clarify the type of meteorological forcing used (e.g., daily vs. smoothed data).

Response: The Sag River basin ATS model applied the daily Daymet while in Teller, the ELM model forcings were based on the 1-hrly ERA5 Land model data. We now have significantly revised and reorganized this section of the paper to be clearer, please see the track-changed version of the manuscript.

**Validation**

Were the models validated using measured temperature, runoff, streamflow, or discharge data? This should be clearly stated.

Response: The ATS simulator's permafrost mode has been compared against laboratory experiments (Painter et al. 2016) and field measurements of temperature, runoff, and hydraulic head in previous studies (e.g., Atchley et al., 2015; Jan et al., 2020; Painter et al. 2023), which lends confidence that our 22 hillslope models are realistic benchmarks for evaluating ELM's runoff parameterization schemes. However, no specific evaluation of ATS was performed for this study.

For ELM, which originates from CLM4.5, the model framework has been continuously developed and validated across numerous studies for hydrological, permafrost, and Arctic applications (e.g., Swenson et al., 2012; Niu et al., 2007; Lawrence et al., 2019). In this study, we applied the optimized coefficients within ELM to simulate runoff at the Teller watershed and directly compared the results against observed streamflow. This dual reliance on a physics-based benchmark (ATS) and independent observations (streamflow at Teller Creek) provides a robust foundation for assessing the performance and generalizability of ELM's runoff parameterization. Importantly, ELM continues to undergo active development, including new capabilities such as a topounit water transfer scheme (e.g., <a href="https://ngee-arctic.ornl.gov/models/integrated-model-2">https://ngee-arctic.ornl.gov/models/integrated-model-2</a> and <a href="https://github.com/E3SM-Project/E3SM/pull/6718">[https://github.com/E3SM-Project/E3SM/pull/6718</a>]), meaning that the evaluation presented here is part of an evolving framework for improving runoff representation in Earth system models.

**Figure-Specific Comments**

**Figure 1:**

- L174: refers to Fig. 1b
- L186: refers to Fig. 1a
- Mark the locations on the Teller map where streamflows were measured.

Response: Thank you for noticing this error. We have corrected it and added the locations of the Teller streamflow gauge to the map.

ATS Model: Clarify whether it was a 2D or full 3D hillslope simulation. If 3D, specify whether runoff refers to streamflow over the entire hillslope.

Response: A hillslope model is constructed as a 2D mesh, including one vertical and one horizontal dimension (i.e., pseudo-3D). These hillslopes were originally generated for the purpose of verifying a novel downscaling modeling conceptualization we proposed for permafrost simulations at a large-scale basin. Full physics permafrost modeling at a large-scale basin usually requires very high computational cost. In our strategy, instead of 3D modeling at a given basin, we propose to apply a series of watershed decomposition and parameterization approaches to construct a bunch of equivalent 2D hillslope models and do full physics simulations at such hillslope levels in parallel and re-construct surface runoff by routing for the whole basin. We have demonstrated that the 2D hillslope models we generated can produce equivalent results, like discharge, with 3D modeling.

In the revised paper, we extensively revied our Methodology Section and added a new subsection (3.2) detailing this information so that is clearer what it was that we did for the ATS modeling design. We additionally updated Figure 2 to be clearer on how we averaged the runoff calculations across the hillslopes.

Figure 2b: Indicate whether discharges are distributed over the hillslope or limited to streamflow.

Response: In ATS, the discharges are distributed over the hillslope. We have modified the figure and caption to reflect this. Please see the track changed version of the manuscript.

Figure 3: Add titles for each subplot column (e.g., Base Case, Adjusted, Seasonally Adjusted). Include "Teller" and "Sag" in the caption.

**Response:** We have updated this figure in the new version of the manuscript. See below the revised figure (Figure R2.3) with changes as suggested, as well as some other minor updates.

Figure R2.3. The revised Figure 3 of the manuscript.

Figure 4: Avoid plotting ATS discharge as dots; use a continuous line.

Response: We replaced the dots with thick continuous lines. Please see the updated figure under the comment below.

Figure 4: Consider plotting the difference (ATS – ELM) to highlight which seasons match better or worse. Use mL instead of 10^-5 units for easier interpretation.

We added a difference plot as a 4th subplot within Figure 4, that has all unadjusted, annually adjusted, and seasonally adjusted scenarios. We spotted a mistake in the printed unit. It is supposed to be kg.m^-2.s^-1, instead of kg.m^-2. We fixed that and converted seconds to days.

Figure R2.4. Revised Figure 4 of the manuscript.

**Discussion Suggestions**

Explicitly use notation such as  $y^{ATS}_{runoff}$  and  $y^{ELM}_{runoff}$  for clarity.

We have now added this notation to the Discussion section. Thank you for this suggestion.

Discuss why RMSE, MAE, and NSE metrics were chosen and what different insights they provide.

Response: In the revised manuscript, we clarified why RMSE, MAE, and NSE were selected and what distinct insights they provide. RMSE is sensitive to large errors and highlights peak

mismatches between simulated and observed runoff, while MAE reflects the average magnitude of deviations regardless of their sign, offering a measure less dominated by outliers. NSE evaluates overall model efficiency by comparing simulation performance to the observed mean, thus indicating how well the model reproduces both the timing and magnitude of streamflow. These clarifications were added to the Section 3.2.

L269: Does the ground refreeze at the Teller site? If not, this may lead to increased water storage and higher subsurface discharge during summer.

Response: The ground does refreeze in the winter. Soils are generally wetter in the winter and drain as they thaw into the spring/summer. Through the warm season, soil moisture is variable. In the cold season, as they freeze up, they hold more water until saturated, with wetter soils higher in the profile, and driest soils at 1.38 m depth. The deeper soil has an attenuated response to soil wetting and drying (i.e. black line below in Figure R2.5). There is an increase in soil moisture during the summer (that late peak that happens mid-summer or towards the end of the summer) in response to precipitation events. But once frozen, soil moisture increases rapidly.

Figure R2.5. Teller27 watershed responses.

Note that ELM is a single grid cell, and there is no heterogeneity across these grid cell itself in this version of the model, thus the responses are the averaged values for the grid cell, which represents a mix of these features.

We now refer specifically to the lack of subgrid variability in the ELM configuration applied in this work on line 249 on the revised paper.

Consider mentioning that Teller's topography, shrub cover, and snow-shrub interception promote warmer soils and influence hydrology.

Response: Our project is working on some improvements to better present snow processes, subgrid snow, and snow-shrub interactions. However, those are not related to the version of the model that we applied in this study. Therefore, we think that the runoff in this iteration of the paper is doing ok given the lack of representation of all of these features. We added a sentence to the paper discussing this [line 922].

L270: Clarify whether the symbol used is a diamond or a triangle.

Response: The diamond symbol is used to represent the ATS's cold season discharge values, while the triangle shows the warm season discharge values in Figure 3 c, f, and i. We revised this figure and updated the visualizations to ensure that the diamonds and the triangles are more clearly distinguished.

Lines 290–295: Include a discussion comparing this study's approach with the CLM group's hillslope paper.

Response: we revised the sentence at lines 290–295 to reference the CLM hillslope framework (Swenson et al., 2019; Lawrence & Swenson, 2024) and added a discussion in Section 5.3 comparing our approach with the CLM representative hillslope method.

**References**

- Atchley, A. L., Painter, S. L., Harp, D. R., Coon, E. T., Wilson, C. J., Liljedahl, A. K., & Romanovsky, V. E. (2015). Using field observations to inform thermal hydrology models of permafrost dynamics with ATS (v0.83). *Geoscientific Model Development*, 8(9), 2701-2722.
- Jan, A., Coon, E. T., & Painter, S. L. (2020). Evaluating integrated surface/subsurface permafrost thermal hydrology models in ATS (v0.88) against observations from a polygonal tundra site. *Geoscientific Model Development*, 13(5), 2259-2276.
- Lawrence, D. M., Fisher, R. A., Koven, C. D., Oleson, K. W., Swenson, S. C., Bonan, G., ... & Zeng, X. (2019). The Community Land Model version 5: Description of new features, benchmarking, and impact of forcing uncertainty. *Journal of Advances in Modeling Earth Systems*, 11(12), 4245-4287.
- Lawrence, D. M., & Swenson, S. C. (2024). Improving terrestrial hydrologic process representation in Earth System Models: Accounting for slope, aspect, and lateral water

- transfer through representative hillslopes. In *AGU Fall Meeting Abstracts* (Vol. 2024, pp. H13R-03).
- Niu, G. Y., Yang, Z. L., Dickinson, R. E., Gulden, L. E., & Su, H. (2007). Development of a simple groundwater model for use in climate models and evaluation with Gravity Recovery and Climate Experiment data. *Journal of Geophysical Research: Atmospheres*, 112(D7): 1-14.
- O'Connor, M. T., Cardenas, M. B., Ferencz, S. B., Wu, Y., Neilson, B. T., Chen, J., and Kling, G. W.: Empirical Models for Predicting Water and Heat Flow Properties of Permafrost Soils, Geophys. Res. Lett., 47, e2020GL087646, https://doi.org/10.1029/2020GL087646, 2020.
- Painter, S. L., Coon, E. T., Khattak, A. J., & Jastrow, J. D. (2023). Drying of tundra landscapes will limit subsidence-induced acceleration of permafrost thaw. *Proceedings of the National Academy of Sciences*, 120(8), e2212171120.
- Swenson, S. C., Lawrence, D. M., & Lee, H. (2012). Improved simulation of the terrestrial hydrological cycle in permafrost regions by the Community Land Model. *Journal of Advances in Modeling Earth Systems*, 4(3), 1-15.